# IMITATION LEARNING FOR MEAN FIELD GAMES WITH CORRELATED EQUILIBRIA

## ABSTRACT

Imitation learning (IL) aims at achieving optimal actions by learning from demonstrated behaviors without knowing the reward function and transition kernels. Conducting IL with a large population of agents is challenging as agents' interactions grow exponentially with respect to the population size. Mean field theory provides an efficient tool to study multi-agent problems by aggregating information on the population level. While the approximation is tractable, it is non-trivial to restore mean field Nash equilibria (MFNE) from demonstrations. Importantly, there are many real-world problems that cannot be explained by the classic MFNE concept; this includes the traffic network equilibrium induced from the public routing recommendations and the pricing equilibrium of goods generated on the E-commerce platform. In both examples, correlation devices are introduced to the equilibrium due to the intervention from the platform. To accommodate this, we propose a novel solution concept named adaptive mean field correlated equilibrium (AMFCE) that generalizes MFNE. On the theory side, we first prove the existence of AMFCE, and establish a novel framework based on IL and AMFCE with entropy regularization (MaxEnt-AMFCE) to recover the AMFCE policy from real-world demonstrations. Signatures from the rough path theory are then applied to characterize the mean-field evolution. A significant benefit of our framework is that it can recover both the equilibrium policy and the correlation device from data. We test our framework against the state-of-the-art IL algorithms for MFGs on several tasks (including a real-world traffic flow prediction problem), results justify the effectiveness of our proposed method and show its potential to predicting and explaining large population behavior under correlated signals.

## 1 INTRODUCTION

Imitation learning (IL) (Hussein et al., 2017) has been widely adopted to learn the desired behavior through expert demonstrations and led to a series of impressive successes (Silver et al., 2016; Shi et al., 2019; Shang et al., 2019). Existing imitation learning algorithms cannot handle tasks with a large group of agents due to the curse of dimensionality and the exponential growth of agent interactions when the number of agents increases. However, many real-world scenarios require the algorithm to handle a large population. Examples include traffic management and control (Bazzan, 2009), Ad auction (Guo et al., 2019), online business with a large customer base (Ahn et al., 2007) and social behaviors between game bots and humans (Jeong et al., 2015). For systems with a large population of homogeneous agents, mean field theory provides a practically efficient and analytically feasible approach to analyze the otherwise challenging multi-agent games (Guo et al., 2019; Yang et al., 2018b). In the mean field game (MFG) setting, the states of the entire population can be sufficiently summarized into an empirical distribution of states thanks to the homogeneity property. Therefore it suffices to consider a game between a representative agent and an empirical distribution.

Existing (and rather limited) literature on mean-field IL assumes that the expert demonstrations are sampled from the classic mean field Nash equilibrium (MFNE) (Yang et al., 2018a; Chen et al., 2022). The limitation of this framework is not general enough to capture many real-world situations where external and correlated signals influence the behavior of the entire populations. Examples include the behavior of drivers on the traffic network with routing recommendations from Google Map or Apple Map. Another possible example is the E-commerce platform recommendation for individual

sellers on setting up the price for their products. In these two examples, a mediator or a coordinator recommends decisions but individual agents who seek for greedy decisions could deviate from the recommendation if she/he finds a better option given the available information. The existence of the mediator introduces correlations among the behaviors of individual agents. Therefore, a more general equilibrium concept is needed before we take a step further to learn from expert demonstrations. Inspired by the concept of correlated equilibrium (CE) for stateless game (Aumann, 1974), there are recent developments on mean field correlated equilibrium (MFCE) with state dynamics. Campi and Fischer assume that a mediator recommends the same stochastic policy to the entire population, resulting in a limited equilibrium set which is the same as the classic MFNE (Campi & Fischer, 2022). In addition, it is often more practical for the mediator to recommend an action rather than a stochastic policy to individuals (see the traffic routing and e-commerce examples). Muller et al. assume that the mediator recommends a time-independent and deterministic policy (sampled from some distribution over the deterministic policy space) to each individual (Muller et al., 2022). This formulation is also rather limited in terms of describing the behaviors of many real-world applications and enabling sufficient flexibility of the population behavior. A more general and practical setting is to establish a framework where the mediator could sample a stochastic policy based on some time-dependent signals and recommend action for each individual, which is the exact framework investigated in this paper. (See Appendix H for a concrete example to show that our equilibrium concept is more general than the one proposed by Muller et al. (Muller et al., 2022).)

Given the above mentioned limitations of current existing MFCE concepts and mean-field IL approaches, we propose a new MFCE framework named adaptive mean field correlated equilibrium (AMFCE) with time-dependent correlated signals and an individual agent can adaptively update her belief on the unobserved correlated signal. We develop a method to recover AMFCE policy based on Maximum Entropy Regularization. Our framework has the following important and novel ingredients:

- **Novel MFCE concept with time-dependent correlated signals and adaptive belief updates from individual agents.** In this paper, we propose a new MFCE framework (called AMFCE) that the mediator recommends an action sampled from a stochastic policy for each agent at every time step. This is a more general and flexible framework compared to previous works on the MFCE (Muller et al., 2022; Campi & Fischer, 2022). We prove the existence of AMFCE solution under mild conditions and prove that MFNE is a subclass of AMFCE.

- **Entropy Regularization to overcome the equilibrium selection difficulty.** Most of the IL algorithms for games face the equilibrium selection issue or identifiability issue as there often exist multiple equilibria. To bypass this difficulty, we further propose an entropy regularized AMFCE (MaxEnt-AMFCE) framework which is shown to have a unique solution.

- **Using signatures from rough path theory to efficiently represent mean-field evolution.** Mean field information is often inaccessible in practice and it is computationally expensive to approximate the mean field information by its empirical distribution. To overcome this difficulty, we adopt signatures from the rough path theory to represent the mean-field evolution, which can be easily combined with neural network training architectures and the resulting method is computationally efficient.

With all these ingredients, our correlated mean field imitation learning (CMFIL) framework can recover not only the policy but also the correlation device, which is the distribution that the correlated signal is sampled from. To the best of our knowledge, this paper is the first focusing on MFCE with the correlation device providing time-dependent recommendations and allowing adaptive belief updates for individual agents.

In addition, we illustrate the performance of our framework by comparing the state-of-the-art imitation learning algorithms for MFGs on several tasks, including a real-world traffic flow prediction problem. The experimental results demonstrate that our framework outperforms the baseline in all tasks. As a by-product, our framework is also suitable for solving MFNE as MFNE is a subclass of AMFCE.

## 2 PRELIMINARY: CLASSIC MEAN FIELD NASH EQUILIBRIUM

This section introduces the classic framework of MFG and the concept of MFNE. The classic MFG models a game between a representative agent and the state distribution of all the other agents.

Denote $\mathcal{P}(\mathcal{X})$ as the set of probability distributions over $\mathcal{X}$ and denote $\mathcal{T} = \{0, 1, \cdots, T\}$ as a set of time indexes. At time $t$, after the representative player chooses her action $a_t$ according to some measurable policy $\pi_t : \mathcal{S} \to \mathcal{P}(\mathcal{A})$, she will receive a deterministic reward $r(s_t, a_t, \mu_t)$ and her state will evolve according to current state $s_t \in \mathcal{S}$ and $P(\cdot|s_t, a_t, \mu_t)$ , where $\mu_t \in \mathcal{P}(\mathcal{S})$ represents the population state distribution and $\mathcal{S}$ is finite. Intuitively, $\mu_t(s) = \lim_{N \to \infty} \frac{\sum_{i=1}^{N} \mathbb{1}_{\{s_t^i = s\}}}{N}$ can be viewed as the limit of the empirical distribution of an homogeneous $N$-agent game where $s_t^i$ is the state of agent $i$ at time $t$ and $\mathbb{1}_{\{e\}}$ is an indicator function (with value 1 if expression $e$ holds and 0 otherwise). Here $P : \mathcal{S} \times \mathcal{A} \times \mathcal{P}(\mathcal{S}) \to \mathcal{P}(\mathcal{S})$.

For fixed mean-field information $\boldsymbol{\mu} = \{\mu_t\}_{t=0}^{T}$, the objective of the representative agent is to solve the following decision-making problem over all admissible policies $\boldsymbol{\pi} = \{\pi_t\}_{t=0}^{T}$:

$$\text{maximize}_{\boldsymbol{\pi}} \quad V_k(s, \boldsymbol{\pi}, \boldsymbol{\mu}) := \mathbb{E}\left[\sum_{t=k}^{T} \gamma^t r(s_t, a_t, \mu_t)\middle| s_k = s\right] \qquad \text{(Classic MFG)}$$
$$\text{subject to} \quad s_{t+1} \sim P(\cdot|s_t, a_t, \mu_t), \quad a_t \sim \pi_t(s_t),$$

The Mean-field Nash Equilibrium (MFNE) is defined as the following.

**Definition 1** (MFNE). *In* (Classic MFG), *a player-population profile* $(\boldsymbol{\pi}^\star, \boldsymbol{\mu}^\star)$ *is called a MFNE (under initial state $\mu_0$) if*

1. *(Single player side) For any policy* $\boldsymbol{\pi}$, *any time index* $t \in \mathcal{T}$, *and any initial state* $s \in \mathcal{S}$, $V_t(s, \boldsymbol{\pi}^\star, \boldsymbol{\mu}^\star) \geq V_t(s, \boldsymbol{\pi}, \boldsymbol{\mu}^\star)$.

2. *(Population side)* $\{\mu_t^*\}_{t=0}^{T}$ *satisfies* $\mu_t^*(\cdot) = \sum_{s \in \mathcal{S}, a \in \mathcal{A}} P(\cdot|s, a, \mu_{t-1}^*)\pi_{t-1}^*(a|s)\mu_{t-1}^*(s)$ *with initial condition* $\mu_0^* = \mu_0$.

The single player side condition captures the optimality of $\boldsymbol{\pi}^\star$, when the population side is fixed. The population side condition ensures the "consistency" of the solution: it guarantees that the state distribution flow of the single player matches the population state sequence $\boldsymbol{\mu}^\star := \{\mu_t^\star\}_{t=0}^{T}$.

## 3 PROBLEM FORMULATION

This section introduces a novel adaptive mean-field correlated equilibrium (AMFCE) framework and establishes the existence of equilibria solutions under mild conditions. We prove that the solution set of AMFCE is richer than the well-known MFNE. Furthermore, the maximum entropy principle is adopted to select the solution with maximum entropy among the solution set of the AMFCE.

### 3.1 ADAPTIVE MEAN FIELD CORRELATED EQUILIBRIUM (AMFCE)

To incorporate the correlations introduced by the central platforms in the traffic network example and the E-commerce marketplace example introduced in Section 1, we consider a mediator (or a central agent) who samples a correlated signal $z_t \in \mathcal{Z}$ at each time $t$, where $\mathcal{Z}$ is a finite signal space. $z_t$ may represent some global conditions such as the weather on day $t$ for the traffic network example and the supply-demand imbalance in month $t$ for the E-commerce marketplace example. Before discussing the AMFCE, we first introduce the concepts of behavioral policy and correlation device.

**Definition 2.** *For each time $t$, the behavioral policy* $\pi_t : \mathcal{Z} \times \mathcal{S} \to \Delta(\mathcal{A})$ *maps from the signal and state spaces to the simplex over the action space. Given the correlated signal $z \in \mathcal{Z}$ and an action $a \sim \pi_t(\cdot|s, z)$ will be independently sampled as a private recommendation for each agent at state $s$.*

**Definition 3.** *The per-step correlation device* $\rho_t \in \Delta(\mathcal{Z})$ *is a publicly known distribution over the space of correlated signal, from which the mediator will sample the correlated signal at time step $t$. Denote* $\boldsymbol{\rho} = \{\rho_t\}_{t=0}^{T}$ *as correlation device over the entire horizon.*

At every time step $t$, a correlated signal $z_t$ is sampled from the per-step correlation device $\rho_t$. Then a recommendation action $a_t$ will be sampled independently from the behavior policy $\pi_t(\cdot|s_t, z_t)$, and sent to each agent at state $s_t$. This recommended action $a_t$ is *private* and only available to the agent. Mathematically, denote $\mathcal{I}_t = \{\rho_t, a_t, \pi_t(\cdot, \cdot, \cdot), s_t, z_{t-1}, \mu_{t-1}\}$ as the information available to the agent at the beginning of step $t$. And $\mathcal{I}_0 = \{\rho_0, a_0, \pi_0(\cdot, \cdot, \cdot), s_0\}$. Note that the agent only observes the functional form of $\pi_t$ but *can not observe* the correlated signal $z_t$ nor the recommended actions for other agents. Based on the information $\mathcal{I}_t$, the agent will take an action $a_t'$ (which may be different from the mediator's recommendation), and then the agent at state $s_t$ will transit to the next state according to distribution $P(\cdot|s_t, a_t', \mu_t) \in \mathcal{P}(\mathcal{S})$ given current mean field $\mu_t$, which follows:

$$\mu_t(\cdot) = \sum_{a \in \mathcal{A}} \sum_{s \in \mathcal{S}} \mu_{t-1}(s) P(\cdot|s, a, \mu_{t-1}) \pi_{t-1}(a|s, z_{t-1}). \tag{1}$$

This implies that, given $\mu_{t-1}$ and $\pi_{t-1}$, $\mu_t$ is fully determined by $z_{t-1}$. After receiving the recommendation action $a_t$, the agent can *predict* the correlated signal by

$$\rho_t^{\mathrm{pred}}(z_t = z|\mathcal{I}_t) = \frac{\rho_t(z)\pi_t(a_t|s_t, z)}{\sum_{z' \in \mathcal{Z}} \rho_t(z')\pi_t(a_t|s_t, z')}. \tag{2}$$

Based on the available information $\mathcal{I}_t$ at time $t$, the agent can then update the prediction on the mean field distribution of the next time-step for each possible signal $z$:

$$\mu_{t+1}^{\mathrm{pred}}(\cdot|\mathcal{I}_t, z) = \sum_{a \in \mathcal{A}} \sum_{s \in \mathcal{S}} \mu_t(s) P(\cdot|s, a, \mu_t) \pi_t(a|s, z) := \Phi(\mu_t, \pi_t, z). \tag{3}$$

The Q function $Q_t^{\boldsymbol{\pi}}(s, a, \mu, z; \boldsymbol{\pi}')$ for individual agent is defined as follows:

$$Q_t^{\boldsymbol{\pi}}(s, a, \mu, z; \boldsymbol{\pi}') = r(s, a, \mu) + \gamma \mathbb{E}_{\boldsymbol{\pi}'}\left[\sum_{i=t+1}^{T} \gamma^{i-t-1} r(s_i, a_i, \mu_i) \bigg| (s_t, a_t, \mu_t, z_t) = (s, a, \mu, z)\right],$$

where $\mathbb{E}_{\boldsymbol{\pi}'}$ is the expectation taken with respect to $z_i \sim \rho_i(\cdot)$, $a_i \sim \pi_i(\cdot|s_i, z_i)$, $s_{i+1} \sim P(\cdot|s_i, a_i, \mu_i)$, $\forall i \in \{t+1, t+2, \cdots, T\}$. We can verify that the Q function satisfies the following Bellman equation:

$$Q_t^{\boldsymbol{\pi}}(s, a, \mu, z; \boldsymbol{\pi}') = r(s, a, \mu) + \gamma \mathbb{E}\left[Q_{t+1}^{\boldsymbol{\pi}}(s', a', \Phi(\mu, \pi_t', z), z'; \boldsymbol{\pi}') \bigg| (s_t, a_t, \mu_t, z_t) = (s, a, \mu, z)\right], \tag{4}$$

where the expectation is taken with respect to $z' \sim \rho_{t+1}(\cdot), s' \sim P(\cdot|s, a, \mu), a' \sim \pi_{t+1}(\cdot|s, z')$.

Similarly, we define the optimal Q-function $Q_t^*(s, a, \mu, z; \boldsymbol{\pi}')$ as the Q function associated with the optimal individual policy $\boldsymbol{\pi}^*$ given population behavior $\boldsymbol{\pi}'$. It is easy to show that $Q^*$ satisfies the following Bellman equation:

$$Q_t^*(s, a, \mu, z; \boldsymbol{\pi}') = r(s, a, \mu) + \gamma \max_{a' \in \mathcal{A}} \mathbb{E}\left[Q_{t+1}^*(s', a', \Phi(\mu, \pi_t', z), z'; \boldsymbol{\pi}') \big| (s_t, a_t, \mu_t, z_t) = (s, a, \mu, z)\right], \tag{5}$$

where the expectation is taken with respect to $z' \sim \rho_{t+1}(\cdot), s' \sim P(\cdot \mid s, a, \mu_t)$.

It is worth noting that if the policy of population $\boldsymbol{\pi}'$ is fixed, $Q_T^*(s, a, \mu, z; \boldsymbol{\pi}') \geq Q_T^{\boldsymbol{\pi}}(s, a, \mu, z; \boldsymbol{\pi}')$ for any $\boldsymbol{\pi}$. Then by induction, it holds that $Q_t^*(s, a, \mu, z; \boldsymbol{\pi}') \geq Q_t^{\boldsymbol{\pi}}(s, a, \mu, z; \boldsymbol{\pi}')$ for all $t \in \mathcal{T}$.

To introduce the concept of AMFCE, we define the set of swap function $\mathcal{U} := \{u : \mathcal{A} \to \mathcal{A}\}$, namely $u$ a function that modifies an action $a$ to an action $u(a)$. Let $\Delta_t(s, \mu, u; \boldsymbol{\pi}, \boldsymbol{\rho}) = \mathbb{E}\left[Q_t^{\boldsymbol{\pi}}(s, u(a), \mu, z; \boldsymbol{\pi}) - Q_t^{\boldsymbol{\pi}}(s, a, \mu, z; \boldsymbol{\pi})\right], u \in \mathcal{U}$ denote the margin of Q function of that agent takes action $u(a)$ when a recommendation $a$ is provided by the mediator, where the expectation is taken with respect to $z \sim \rho_t(\cdot), a \sim \pi_t(\cdot|s, z)$.

**Definition 4.** *The profile $(\boldsymbol{\pi}^\star, \boldsymbol{\rho})$ composed of the time-varying stochastic policy $\boldsymbol{\pi}^\star = \{\pi_t^\star\}_{t=0}^T$ and the correlation device $\boldsymbol{\rho}$ is an adaptive mean field correlated equilibrium (AMFCE) if*

- *(Single agent side) No agent has an incentive to unilaterally deviate from the recommendation action after predicting the $z$ by (2), i.e.*

$$\Delta_t(s, \mu_t^\star, u; \boldsymbol{\pi}^\star, \boldsymbol{\rho}) \leq 0, \quad \forall u \in \mathcal{U}, \forall s \in \mathcal{S}, \forall t \in \mathcal{T}.$$

| Equilibrium | MFNE | | | AMFCE | | | |
|---|---|---|---|---|---|---|---|
| Distribution | $\pi_0^0(a=L\|s=\cdot)$ | $\pi_0^1(a=L\|s=\cdot)$ | $\pi_0^2(a=L\|s=\cdot)$ | $\pi_0(a=L\|s=\cdot,z=0)$ | $\pi_0(a=L\|s=\cdot,z=1)$ | $\rho_0(z=0)$ | $\rho_0(z=1)$ |
| Value | 1 | 0 | 1/2 | 2/3 | 1/3 | 1/2 | 1/2 |

Table 1: AMFCE and MFNE in the Ocean Ranch. The AMFCE shown in this table is not an MFNE.

- *(Population side) $\{\mu_t^*\}_{t=0}^T$ satisfies $\mu_t^*(\cdot) = \sum_{s\in\mathcal{S}, a\in\mathcal{A}} P(\cdot|s,a,\mu_{t-1}^*)\pi_{t-1}^*(a|s,z_{t-1})\mu_{t-1}^*(s)$ given the correlated signals $\{z_t\}_{t=0}^T$ and with initial condition $\mu_0^* = \mu_0$.*

A toy example named Ocean Ranch is provided below to demonstrate the concept of AMFCE.

**Example 1.** *Suppose there exists a marine ranch with two sectors. The regulator of the marine ranch adjusts the size of the fish entering the two different sectors by giving recommendations for fish. The state space of fish is $\mathcal{S} = \{C, L, R\}$, and the action space is $\mathcal{A} = \{L, R\}$. Initial mean field $\mu_0(C) = 1$. The reward $r(s,a,\mu) = \mathbb{1}_{\{s=L\}}\mu(L) + \mathbb{1}_{\{s=R\}}\mu(R)$ and $\mathcal{T} = \{0, 1\}$. The environment dynamic is deterministic: $P(s_{t+1} = R \mid s_t = \cdot, a = R) = 1$, $P(s_{t+1} = L \mid s_t = \cdot, a = R) = 0$, $P(s_{t+1} = R \mid s_t = \cdot, a = L) = 0$, $P(s_{t+1} = L \mid s_t = \cdot, a = L) = 1$.*

We prove that in Example 1, the regulator in an AMFCE gives recommendations as follows (see the detailed proof in Appendix C). First, a random variable $z$ is sampled from the correlated signal space $\mathcal{Z} = \{0, 1\}$ with equal probability $\rho_0(z=0) = \rho_0(z=1) = 0.5$, and the regulator gives the action recommendation for each fish according to the policy $\pi_0(a=L|z=0) = 2/3$, $\pi_0(a=R|z=0) = 1/3$, $\pi_0(a=L|z=1) = 1/3$, $\pi_0(a=R|z=1) = 2/3$. Then fish has no incentive to deviate from the recommendation, so an AMFCE is achieved. It is worth noting that the above AMFCE solution is **not** a classic MFNE, because there are only three MFNEs which are shown in Table 1.

### 3.2 PROPERTIES OF AMFCE

This section focuses on the properties of AMFCE, including the conditions to guarantee the existence and its relationship to classic MFNE.

In order to provide the existence of AMFCE solutions, we define the best response operator $\mathrm{BR}(\boldsymbol{\pi}; \boldsymbol{\rho}) = \arg\max_{\boldsymbol{\pi}'} \mathbb{E}_{\boldsymbol{\pi}', \boldsymbol{\rho}}\left[\sum_{t=0}^T \gamma^t r(s_t, a_t, \mu_t)\right]$, where the expecation is taken with respect to $z_t \sim \rho_t(\cdot), s_t \sim P(\cdot|s_{t-1}, a_{t-1}, \mu_{t-1}), a_t \sim \pi_t'(\cdot|s_t, z_t), \mu_t = \Phi(\mu_{t-1}, \pi_{t-1}, z_{t-1})$. Unless otherwise stated, the expectation $\mathbb{E}_{\boldsymbol{\pi}, \boldsymbol{\rho}}$ is taken with respect to $z_t \sim \rho_t(\cdot), s_t \sim P(\cdot|s_{t-1}, a_{t-1}, \mu_{t-1}), a_t \sim \pi_t(\cdot|s_t, z_t), \mu_t = \Phi(\mu_{t-1}, \pi_{t-1}, z_{t-1})$. Then the existence of the solution will be derived using Kakutani's fixed point theorem (Kakutani, 1941) with the operator BR. We next provide a sufficient condition for the existence of AMFCE.

**Theorem 1.** *If the functions $r(s, a, \mu)$ and $P(s'|s, a, \mu)$ are bounded and continuous with respect to $\mu$, there exists an AMFCE solution.*

The AMFCE is a more general equilibrium concept than MFNE, which is illustrated in corollary 1.

**Corollary 1.** *If $(\boldsymbol{\pi}, \boldsymbol{\mu})$ is an MFNE, then it leads to an AMFCE solution $(\boldsymbol{\pi}, \boldsymbol{\rho})$ with $|\mathcal{Z}| = 1$ and $\rho_t(z) = 1$ for all $t \in \mathcal{T}$ where $z \in \mathcal{Z}$ is the single element in the signal space.*

The proof is deferred to Appendix D.3. This proposition implies that the MFNE is a subset of AMFCE. The example in Example 1 shows that AMFCE may not be an MFNE.

### 3.3 MAXIMUM ENTROPY MEAN FIELD CORRELATED EQUILIBRIUM

Similar to the classic MFG setting, there may be multiple AMFCEs in our setting. Consequently, AMFCE is facing the equilibrium selection issue. One of the commonly used selection criteria is maximum entropy. For example, maximum entropy has been introduced to select correlated equilibrium in the normal form game (Ortiz et al., 2007) and Markov game (Ziebart et al., 2011). We integrate the maximum entropy principle into the AMFCE as follows.

**Definition 5.** *The maximum entropy AMFCE (MaxEnt-AMFCE) is the one that maximizes the entropy $(\boldsymbol{\pi}^*, \boldsymbol{\rho}^*) = \arg\max_{(\boldsymbol{\pi}, \boldsymbol{\rho}) \in \Pi_{\mathrm{AMFCE}}} H(\boldsymbol{\pi}, \boldsymbol{\rho})$, with $H(\boldsymbol{\pi}, \boldsymbol{\rho}) = \sum_{t=0}^T \mathbb{E}_{\boldsymbol{\pi}, \boldsymbol{\rho}}[-\log(\pi_t(a_t|s_t, z_t)\rho_t(z_t))]$, $\Pi_{\mathrm{AMFCE}}$ the set of all AMFCE solutions.*

The MaxEnt-AMFCE can avoid the equilibrium selection problem as it is unique under certain conditions. Denote $\Delta(\boldsymbol{\pi}, \boldsymbol{\rho}) = \max_{u,s,t} \Delta_t(s, \mu_t, u; \boldsymbol{\pi}, \boldsymbol{\rho})$, where $\mu_t = \Phi(\mu_{t-1}, \pi_{t-1}, z_{t-1})$.

**Corollary 2.** *MaxEnt-AMFCE is a unique equilibrium solution if $\Delta(\boldsymbol{\pi}, \boldsymbol{\rho})$ is convex w.r.t. $(\boldsymbol{\pi}, \boldsymbol{\rho})$.*

The proof is deferred to Appendix D.4. Directly optimizing the entropy is difficult because the policy $\pi_t$ and $\rho$ are coupled. So we decouple this term by the following proposition.

**Proposition 1.** *The entropy can be decoupled: $H(\boldsymbol{\pi}, \boldsymbol{\rho}) = \sum_{t=0}^{T} [H(\rho_t) + \mathbb{E}_{\boldsymbol{\pi}, \boldsymbol{\rho}} H(\pi_t|s_t, z_t)]$.*

$H(\rho_t)$ is the entropy of the correlation device, and $H(\pi_t|s_t, z_t) = -\sum_{a_t \in \mathcal{A}} \pi_t(a_t|s_t, z_t) \log(\pi_t(a_t|s_t, z_t))$ is the entropy of $\pi_t(\cdot|s_t, z_t)$. (See proof in Appendix D.5).

## 4 IMITATION LEARNING FOR MEAN FIELD GAME

This section proposes a new framework based on imitation learning to recover AMFCE from collected expert demonstrations. To avoid the equilibrium selection problem, we choose the MaxEnt-AMFCE solution introduced in Section 3.3.

To emphasize the role of unknown reward function in imitation learning, we use $\mathrm{MFRL}(r, \boldsymbol{\rho})$ to denote the policy of MaxEnt-AMFCE under the reward function $r$ and correlation device $\boldsymbol{\rho}$:

$$\mathrm{MFRL}(r, \boldsymbol{\rho}) = \underset{\substack{\boldsymbol{\pi} \\ (\boldsymbol{\pi}, \boldsymbol{\rho}) \in \Pi_{\mathrm{AMFCE}}}}{\arg\min} \ \alpha \sum_{t=0}^{T} \mathbb{E} H(\pi_t|s_t, z_t) \tag{6}$$

The temperature constant $\alpha \geq 0$ is to control the entropy. The constraint on the AMFCE set makes the optimization problem (6) challenging. To address this, we provide an equivalent formulation in Proposition 2 and derive a Lagrangian reformulation of (6).

### 4.1 CORRELATED MEAN FIELD IMITATION LEARNING

We denote $J(\boldsymbol{\pi}, \boldsymbol{\rho}) = \mathbb{E}\left[\sum_{t=0}^{T} \gamma^t r(s_t, a_t, \mu_t)\right]$, and $\mathcal{R}(a_{0:T}, \boldsymbol{\pi}, \boldsymbol{\rho})$ as the margin of expected return between choosing actions $a_{0:T} := \{a_t\}_{t \in \mathcal{T}}$ and policy $\boldsymbol{\pi}$ under the correlation device $\boldsymbol{\rho}$: $\mathcal{R}(a_{0:T}, \boldsymbol{\pi}, \boldsymbol{\rho}) \triangleq \mathbb{E}\left[\sum_{t=0}^{T} \gamma^t r(s_t, a_t, \mu_t)\big|a_{0:T}\right] - J(\boldsymbol{\pi}, \boldsymbol{\rho})$, where the expectation is taken with respect to $z_t \sim \rho_t(\cdot)$, $s_t \sim P(\cdot|a_{t-1}, s_{t-1}, \mu_{t-1})$. And $\mu_t = \Phi(\mu_{t-1}, \pi_{t-1}, z_{t-1})$. Then we can get an equivalent constraint of AMFCE.

**Proposition 2.** $(\boldsymbol{\pi}, \boldsymbol{\rho})$ *is an AMFCE solution if and only if $\mathcal{R}(a_{0:T}, \boldsymbol{\pi}, \boldsymbol{\rho}) \leq 0, \forall a_{0:T} \in \mathcal{A}^{\mathcal{T}}$.*

The proof is deferred to Appendix D.6. Compared to the original formulation (6), it is easier to work with a dual representation without constraints:

$$L(\boldsymbol{\pi}, \boldsymbol{\rho}, \lambda, r) \triangleq \sum_{\tau_k \in \mathcal{D}_E} \lambda(\tau_k) \left( \mathbb{E}\left[\sum_{t=0}^{T} \gamma^t r(s_t, a_t, \mu_t)\right] - J(\boldsymbol{\pi}, \boldsymbol{\rho}) \right) - \alpha \sum_{t=0}^{T} \mathbb{E} H(\pi_t|s_t, z_t) \tag{7}$$

where $\mathcal{D}_E$ is a set of action-signal sequence $\tau_k = \{a_0, z_0, a_1, z_1, a_2, z_2, \cdots, a_T, z_T\}$. We show that (7) captures the difference of expected returns between two policies by selecting $\lambda$ as follows.

**Theorem 2.** *For policy $\boldsymbol{\pi}$ and correlation device $\boldsymbol{\rho}$, let $\lambda_{\boldsymbol{\pi}}^*(\tau_k) = \prod_{t=0}^{T} \rho_t(z_t) \pi_t^*(a_t|s_t, z_t)$ be the probability of generating the sequence $\tau_k$ if the individual policy is $\boldsymbol{\pi}^*$. Then we have $L(\boldsymbol{\pi}, \boldsymbol{\rho}, \lambda_{\boldsymbol{\pi}}^*, r) = \mathbb{E}[\sum_{t=0}^{T} \gamma^t r(s_t, a_t, \mu_t)] - J(\boldsymbol{\pi}, \boldsymbol{\rho}) - \alpha \sum_{t=0}^{T} \mathbb{E}_{\boldsymbol{\pi}, \boldsymbol{\rho}} H(\pi_t|s_t, z_t)$, where the expectation is taken with respect to $z_t \sim \rho_t(\cdot)$, $s_t \sim P(\cdot|s_{t-1}, a_{t-1}, \mu_{t-1})$, $a_t \sim \pi_t^*(\cdot|s_t, z_t)$, $\mu_t = \Phi(\mu_{t-1}, \pi_{t-1}, z_{t-1})$.*

The proof of Theorem 2 is deferred to Appendix D.7.

In the setting of imitation learning, the reward signal is not accessible. To construct a suitable reward function rationalizing the expert policy, we need to define a suitable AMFCE inverse reinforcement learning (AMFCE-IRL) operator which designs a reward to maximize the margin of expected return between expert policy and the other policies:

$$\mathrm{AMFCE\text{-}IRL}_\psi(\boldsymbol{\pi}^E, \boldsymbol{\rho}^E) = \arg\max_r \left( -\psi(r) - \max_{\boldsymbol{\pi}} L(\boldsymbol{\pi}^E, \boldsymbol{\rho}^E, \lambda_{\boldsymbol{\pi}}^*, r) \right), \tag{8}$$

---

**Algorithm 1:** Correlated mean field imitation learning (CMFIL)

---

**Data:** Expert trajectories $\mathcal{D}_E = \{s_0, z_0, a_0, s_1, z_1, a_1, \ldots s_T, z_T, a_T\}$ Initial mean field $\mu_0$, The
      weight of gradient penalty $\beta$
**Result:** Policy $\boldsymbol{\pi}^\theta$, correlation device $\boldsymbol{\pi}^\phi$
Initialization the parameter $\theta$ of policy $\boldsymbol{\pi}^\theta$ and the parameter $\phi$ of correlation device $\boldsymbol{\rho}^\phi$;
**for** *each iteration* **do**
    Obtain trajectories from $(\boldsymbol{\pi}, \boldsymbol{\rho})$ by the process:
    $s_0 \sim \mu_0, a_t \sim \pi^\theta(\cdot|s_t, z_t), s_{t+1} \sim P(\cdot \mid s_t, \mu_t), z_t \sim \rho_t^\phi(\cdot)$;
    Approximate $\mu_t$ with the signature $\hat{\mu}_t = S(\{z_i\}_{i=0}^t)$ using (11);
    **for** *i in* $\{0, 1, 2, \ldots\}$ **do**
        Update $\omega$ to increase the objective

$$\mathbb{E}_{\boldsymbol{\pi}, \boldsymbol{\rho}^E}\Big[\sum_{t=0}^T \gamma^t \log D_\omega(s_t, a_t, \hat{\mu}_t)\Big] + \mathbb{E}_{\boldsymbol{\pi}^E, \boldsymbol{\rho}^E}\Big[\sum_{t=0}^T \gamma^t \log\big(1 - D_\omega(s_t, a_t, \hat{\mu}_t)\big)\Big]$$

    **end**
    **for** *t in* $\{0, 1, 2, \ldots\}$ **do**
        Update $\theta$ by SAC with small step size:

$$\mathbb{E}\Big[\nabla_\theta \rho_t^\phi(z_t)\pi_t^\theta(a_t|s_t, z_t)Q_t^{\boldsymbol{\pi}^\theta}(s_t, a_t, \hat{\mu}_t, z_t; \boldsymbol{\pi}) + \alpha \nabla_\theta H(\pi_t^\theta|s_t, z_t)\Big]$$

        where the expectation is taken with respect to $s_0 \sim \mu_0, a_t \sim \pi^\theta(\cdot|s_t, z_t)$,
        $s_{t+1} \sim P(\cdot \mid s_t, \mu_t), z_t \sim \rho_t^\phi(\cdot)$;
        Update $\phi$ with (10);
    **end**
**end**

---

$(\boldsymbol{\pi}^E, \boldsymbol{\rho}^E) \in \Pi_{\text{MaxEnt-AMFCE}}$ is the MaxEnt-AMFCE from which expert demonstrations are sampled.
We choose a special regularizer (Ho & Ermon, 2016):

$$\psi_{GA}(r) \triangleq \begin{cases} \mathbb{E}[\sum_{t=0}^T \gamma^t g(r(s_t, a_t, \mu_t))] & \text{if } r > 0 \\ +\infty & \text{otherwise} \end{cases} \quad, \quad \text{where} \quad g(x) = \begin{cases} x - \log(1 - e^{-x}) & \text{if } x > 0 \\ +\infty & \text{otherwise} \end{cases}$$

After getting the reward function $\tilde{r} = \text{AMFCR-IRL}(\boldsymbol{\pi}^E, \boldsymbol{\rho}^E)$, we can characterize the AMFCE
policy $\text{MFRL}(\tilde{r}, \boldsymbol{\rho}^E)$ with the learned $\tilde{r}$.

**Proposition 3.** *The policy $\boldsymbol{\pi}$ learned on the reward function recovered by AMFCE-IRL can be characterized as follows:*

$$\text{MFRL} \circ \text{AMFCE-IRL}_\psi(\boldsymbol{\pi}^E, \boldsymbol{\rho}^E) := \quad \arg\min_{\boldsymbol{\pi}} \max_r J(\boldsymbol{\pi}^E, \boldsymbol{\rho}^E) - \mathbb{E}[\sum_{t=0}^T \gamma^t r(s_t, a_t, \mu_t)] - \psi_{GA}(r)$$

*where the expectation is taken with respect to $z_t \sim \rho_t^E(\cdot)$, $s_t \sim P(\cdot|s_{t-1}, a_{t-1}, \mu_{t-1})$, $a_t \sim \pi_t(\cdot|s_t, z_t)$, $\mu_t = \Phi(\mu_{t-1}, \pi_{t-1}^E, z_{t-1})$.*

*The objective to recover MaxEnt-AMFCE is defined as:*

$$\min_{\boldsymbol{\pi}} \max_\omega \mathbb{E}_{\boldsymbol{\pi}, \boldsymbol{\rho}^E}\Big[\sum_{t=0}^T \gamma^t \log D_\omega(s_t, a_t, \mu_t)\Big] + \mathbb{E}_{\boldsymbol{\pi}^E, \boldsymbol{\rho}^E}\Big[\sum_{t=0}^T \gamma^t \log\big(1 - D_\omega(s_t, a_t, \mu_t)\big)\Big] \quad (9)$$

*where $D_\omega$ is the discriminator network parameterized with $\omega$, with input $(s_t, a_t, \mu_t)$ and output a real number in $(0, 1]$. The first expectation is taken with respect to $z_t \sim \rho_t^E(\cdot)$, $s_t \sim P(\cdot|s_{t-1}, a_{t-1}, \mu_{t-1})$, $a_t \sim \pi_t(\cdot|s_t, z_t)$, $\mu_t = \Phi(\mu_{t-1}, \pi_{t-1}^E, z_{t-1})$.*

The proof is deferred to Appendix D.8. From a theoretical point of view, we assume that neural
network $D_\omega$ has the capacity to approximate the reward function. Under this assumption, the AMFCE
$(\boldsymbol{\pi}^E, \boldsymbol{\rho}^E)$ could be recovered by optimizing the above objective (9). Note that simply applying GAIL
to solve AMFCE cannot recover $\boldsymbol{\rho}^E$, so we derive $\boldsymbol{\rho}$ using a gradient descent method (with proof in
Appendix D.9):

| Experiment | Metric | CMFIL | MFIRL | Logistic Regression | Multinomial | MaxEnt ICE |
|---|---|---|---|---|---|---|
| Squeeze with $T = \{0,1,2\}$ | Log Loss($\pi_0(\cdot \mid z=0)$) | **0.643 (0.000)** | 1.450 (2.857) | 4.484 (0.054) | 0.686 (0.002) | - |
| | Log Loss($\pi_0(\cdot \mid z=1)$) | 0.647 (0.003) | 3.245 (1.650) | **0.000 (0.000)** | 2.577 (0.149) | - |
| | Log Loss($\pi_1(\cdot \mid z=0)$) | **0.020 (0.001)** | 1.072 (2.229) | 7.091 (0.107) | 0.282 (0.087) | - |
| | Log Loss($\pi_1(\cdot \mid z=1)$) | 0.045 (0.005) | 7.871 (4.368) | 10.638 (0.163) | **0.001 (0.001)** | - |
| Squeeze with $T = \{0,1\}$ | Log Loss($\pi(\cdot \mid z=0)$) | **0.648 (0.002)** | 3.828 (1.582) | 1.985 (0.165) | 0.991 (0.102) | 0.946 (0.073) |
| | Log Loss($\pi(\cdot \mid z=1)$) | **0.638 (0.001)** | 2.009 (1.191) | 2.139 (0.169) | 2.947 (0.359) | 0.648 (0.011) |
| RPS | Log Loss($\pi$) | **1.083 (0.000)** | 7.127 (0.753) | 4.805 (0.131) | 5.850 (0.306) | 1.537 (0.019) |
| Flock | Log Loss($\pi(\cdot \mid s=\cdot, z=0)$) | 0.002 (0.000) | 5.591 (0.869) | **0.000 (0.000)** | 1.383 (0.004) | - |
| | Log Loss($\pi(\cdot \mid s=\cdot, z=1)$) | **0.016 (0.003)** | 11.687 (1.158) | 7.887 (0.031) | 1.127 (0.007) | - |
| | Log Loss($\pi(\cdot \mid s=\cdot, z=2)$) | **0.045 (0.009)** | 7.500 (3.955) | 18.339 (0.010) | 0.951 (0.009) | - |
| | Log Loss($\pi(\cdot \mid s=\cdot, z=4)$) | **0.026 (0.003)** | 3.847 (3.967) | 35.253 (0.037) | 1.264 (0.011) | - |

Table 2: Results for numerical tasks.

| | Lewisham | Hammersmith | Ealing | Redbridge | Enfield | Big Ben |
|---|---|---|---|---|---|---|
| CMFIL | **0.742 (0.011)** | **0.897 (0.002)** | **1.091 (0.001)** | **0.052 (0.011)** | **0.394 (0.003)** | **1.599 (0.000)** |
| MFIRL | 12.346 (0.294) | 9.853 (2.892) | 11.625 (0.435) | 11.720 (0.633) | 11.750 (0.603) | 7.482 (1.539) |

Table 3: The results of predicted traffic flow for Traffic Network.

**Proposition 4.** *If $\boldsymbol{\rho}^\phi$ is parameterized with $\phi$, the gradient to optimize $\phi$ given state $s$ is*

$$\mathbb{E}_{z \sim \rho_t^\phi(\cdot)}\left[\nabla_\phi \log \rho_t^\phi(z)\left(-\alpha \log \rho_t^\phi(z) + \alpha H(\pi_t(a|s,z)) + \mathbb{E}_{a \sim \pi_t(\cdot|s,z)} Q_t^{\boldsymbol{\pi}}(s,a,\mu,z;\boldsymbol{\pi})\right)\right].$$

(10)

Now we propose the imitation learning algorithm for AMFCE (Algorithm 1). It is worth noting that this algorithm can recover AMFCE that does not have the maximum entropy by setting $\alpha = 0$.

### 4.2 REPRESENTATION OF THE MEAN FIELD INFORMATION

As the mean field appears in the input of discriminator $D_\omega(s, a, \mu)$ in (9), it is necessary to find an efficient way to represent the mean field information.

In the Kolmogorov equation (1), the mean field flow $\{\mu_t\}_{t=0}^{T}$ is deterministic given fixed correlated signal sequence $\{z_t\}_{t=0}^{T}$ and given the initial mean field distribution $\mu_0$. Therefore, the mean field distribution $\mu_t$ can be characterized by $\boldsymbol{z}_{0:t} = \{z_i\}_{i=0}^{t}$. Motivated by this, we use the signatures of $\boldsymbol{z}_{0:t}$ from the rough path theory (Kidger & Lyons, 2021; Min & Hu, 2021) to represent the signal sequence and hence to characterize the mean field flow with $\hat{\mu}_t = S(\boldsymbol{z}_{0:t})$. The signatures provide a graduated summary of the path $\boldsymbol{z}_{0:t}$. Therefore, the input of discriminator $D_\omega$ in (9) could be replaced with $(s_t, a_t, \hat{\mu}_t)$. It is worth noting that the signature has been recently applied to the field of machine learning to extract characteristic features of sequential data in a non-parametric fashion (Min & Ichiba, 2020; Ni et al., 2020). The use of signatures to encode historical information avoids heavy computational load which often suffered in tasks like training recurrent neural networks. In addition, the training stability can be significantly enhanced since the mapping is invariant.

**Definition 6.** *Let $\mathbf{x} = \{x_1, \ldots, x_L\}$ with $x_i \in \mathbb{R}^d$, for all $i$ and $L \geq 2$. Denote $f : [0,1] \to \mathbb{R}^d$ to be the continuous piecewise affine function such that $f(\frac{i-1}{L-1}) = x_i$, $\forall i \in \{1, 2, \ldots, L\}$.*

$$S(f)_{0,1} = (1, M_1, \cdots, M_n, \ldots)$$

(11)

*where $M_n = \int_{s < s_1 < \cdots < s_n < t} \frac{df}{dt}(s_1) \otimes \cdots \otimes \frac{df}{dt}(s_n) dt_1 \cdots dt_n$.*

*The signature of the path $\mathbf{x}$ is defined to be $S(f)_{0,1}$, denoted as $S(\mathbf{x})$.*

Signature of sequential data includes infinite terms as shown in the (11), but fortunately, terms $M_n$ enjoy factorial decay. In practice we select the first $n$ terms of the signature without losing crucial information of the data (Kidger et al., 2019).

## 5 EXPERIMENTS

We evaluate the effectiveness of our algorithm in four environments: Sequential Squeeze, RPS, Flock, and Traffic Flow Prediction. We compare our CMFIL framework with MFIRL (Chen et al.,

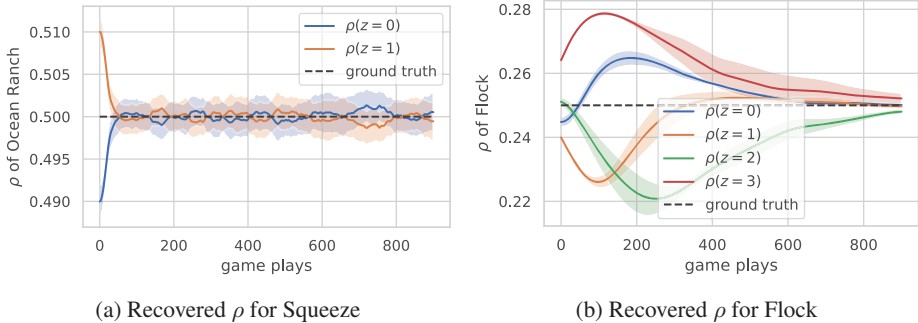

(a) Recovered $\rho$ for Squeeze

(b) Recovered $\rho$ for Flock

Figure 1: The distribution of correlation device $\rho$ recovered by CMFIL.

2021), as it is so far the only method to solve MFNE without requiring the knowledge on the reward. Since MFIRL does not consider correlated signals, we regard the correlated signal as an extension of the global state for their framework. We also compare CMFIL with MaxEnt ICE, smoothed multinomial distribution over the joint actions and logistic regression (Waugh et al., 2013). As MaxEnt ICE is designed to recover correlated equilibrium in matrix game, we only compare CMFIL with MaxEnt ICE on tasks that can be reduced to matrix game, such as RPS and Sequential Squeeze with $\mathcal{T} = \{0, 1\}$. We use the log loss, $\mathbb{E}_{a \sim \pi(\cdot|s,z)}[- \log(\hat{\pi}(a|s,z))]$, to mearsure the difference over recovered policy $\hat{\pi}$ and ground truth $\pi$. The Appendix F contains more details.

We evaluate CMFIL on several tasks: Sequential Squeeze (Squeeze for short), Rock-Paper-Scissors (RPS), Flock and a real-world traffic flow prediction task. The first three experiments are numerical experiments. The traffic flow prediction task is to predict the traffic flow a complex traffic network based on the real world data. Details are presented in the Appendix E.

**Squeeze:** Sequential Squeeze is a game with multi-steps. The purpose to implement this game is to verify the ability to recover expert policy through demonstrations sampled from a multi-step game. The learning curve is shown in the Fig.3, and the results are shown in Table 2. The example of Ocean Ranch in Example 1 is a special case of Sequential Squeeze, where the horizon equals to 2.

**RPS:** This experiment is a traditional mean field game task (Chen et al., 2021; Cui & Koeppl, 2021). The demonstrations are sampled from MFNE, and the cardinality of the correlated signal set is one. We use RPS to verify that the algorithm proposed can recover the expert demonstrations sampled from MFNE, which also supports the results in Corollary 1.

**Flock:** The experiment is based on the movement of fish (Perrin et al., 2021). In nature, fish spontaneously aligns velocity according to the overall movement of the fish school, so that the final fish school forms a stable movement velocity. The video provided shows the convergence process (https://sites.google.com/view/mean-field-imitation-learning/).

**Traffic Flow Prediction:** In the Traffic Flow Prediction task, we use the traffic data of London from Uber Movement. The goal of this experiment is to predict the traffic flow of a traffic network (with six locations) in real-world. Given the large-scale and high-complexity of this task, we compare CMFIL and MFIRL under this task to test their scalability.

The results for numerical tasks are shown in Table 2. CMFIL is better than other methods in general. Supervised learning methods such as logistic regression and smoothed multinomial distribution easily overfit. They may outperform CMFIL in some metrics but suffer from a higher loss than CMFIL in general. MFIRL shows larger deviations and higher loss than CMFIL in Table 2 and Table 3. The reason is that MFIRL can not recover AMFCE, and it can not handle correlated signals properly. Although we have regarded correlated signal as an extension of state. The reward recovered by MFIRL is biased because the ground truth reward is independent of the correlated signal. Furthermore, CMFIL adds a regularizer $\psi$ for the reward function to avoid overfitting, so it also outperforms MFIRL in RPS in which expert demonstrations are sampled from MFNE. MaxEnt ICE also performs poorly because it has a limited reward function class by assuming a linear reward structure. Figure 1 shows that CMFIL can recover the correlation device with a fast convergence speed.

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
