# OpenReview forum: "Imitation Learning for Mean Field Games with Correlated Equilibria"
_ICLR.cc/2023/Conference — Submitted to ICLR 2023_

### Official Review · Reviewer_a1Rt · 2022-10-23

**Confidence:** 2
**Correctness:** 4
**Technical Novelty And Significance:** 3
**Empirical Novelty And Significance:** Not applicable
**Recommendation:** 6

**Clarity, Quality, Novelty And Reproducibility:**

**Clarity:**
- Equations, such as (4) and (5), are formatted without spaces after the equal signs

**Other Questions:**
- It seems like your algorithm finds the maximum entropy mean field correlated equilibrium instead of the AMFCE, and I am wondering how is the max entropy version related to the adapted version of the equilibrium. Especially, which equilibrium does it converge to when the regularization goes to zero?


**Strength And Weaknesses:**

**Strength:**
- The model considers a generalized version of Nash equilibrium where the recommendations are sampled from a stochastic policy. This is a more general framework compared to the existing work.
- The authors provided an imitation learning algorithm that finds the unique equilibrium of the problem.
- The computational efficiency of the mean-field states is enhanced with the introduction of the signatures and the neural network training architectures.
- There are numerical results provided that compare with some of the state-of-the-art imitation learning algorithms and showcase the effectiveness of the proposed algorithm.

**Weakness:**
- There are very few words on the priliminaries of imitation learning, and the introduction of the algorithm CMFIL is not well arranged from my perspective. They are less friendly to readers without enough background knowledge on such topics.
- Section 4.2 seems to be too brief to me, and it is not fully explained how the signature is used for training.

**Summary Of The Paper:**

The paper provides an imitation learning framework that solves correlated equilibria of mean-field games. The authors proposed an adaptive mean field correlated equilibrium as a generalization of the classic mean field Nash equilibrium that considers the correlated devices. They also offered a framework for imitation learning with the help of entropy regularization.

**Summary Of The Review:**

I think the authors considered a fairly interesting problem setting and proposed a generalization of Nash equilibrium for mean-field games. The proposed algorithm for solving the max entropy equilibrium also enjoys good empirical performance on some of the real-world datasets. Despite some of the clarity issues regarding how the paper is written, I feel like it is a solid paper on learning mean-field games.

==========================

I thank the reviewers for addressing my concerns. I will keep my score for the paper.

---

> ### Author Response · Authors · 2022-11-18
> **Response**
>
> Thank you for your positive feedback! We have responded below to each of your comments in detail. All major changes of the submission are marked in red.
> > The introduction to IL and Section 4.2 are too short.
>
> Thank you for the comments. We would love to elaborate on these two parts. We have added some detailed introductions to IL in Appendix B. And we have further explained how the signature is used for training in Section 4.2.
>
>
> >I am wondering how is the max entropy version related to the adapted version of the equilibrium.
>
> All the adaptive mean field correlated equilibria (AMFCE) that satisfy Definition 4 are adaptive. In particular, MaxEnt-AMFCE is a subclass of AMFCE that have the maximum entropy among all the AMFCE solutions. Note that the MaxEnt-AMFCE is unique if the condition in Corollary 2 is further satisfied.
>
>  We assume that the demonstrations are sampled from the MaxEnt-AMFCE. This is because equilibrium with bigger entropy is expected to be more stable in complex stochastic systems and have better exploration structure. We want to emphasize that this assumption is not necessary and our algorithm can, indeed, recover any AMFCE.
>
> >Which equilibrium does it converge to when the regularization goes to zero?
>
> There are two parts related to entropy regularization in our paper. The first part is the entropy of the AMFCE from which we collect expert demonstrations. The second part is the entropy in equation (7).
>
> $L(\pmb{\pi}, \pmb{\rho}, \lambda, r) \triangleq  - \alpha\sum_{t=0}^{T}\mathbb{E}H(\pi_t|s_t, z_t)$
> $+\sum_{\tau_k \in \mathcal{D}_E}\lambda(\tau_k)\mathcal{R}(\pmb{a})\quad(7)$,
>
> where $\mathcal{R}(\pmb{a})=\mathbb{E}[\sum_{t=0}^{T}\gamma^t r(s_t, a_t, \mu_t)] - J(\pmb{\pi}, \pmb{\rho})$, $\pmb{a}=$ {$a_t$}$_{t=0}^{T}$.
>
>
> 1. **The first part**: As the MaxEnt-AMFCE has better exploration  and better stability properties, we assume that the demonstrations are sampled from a MaxEnt-AMFCE, denoted as $(\pmb{\pi}^{E}, \pmb{\rho}^{E})$. For this given MaxEnt-AMFCE $(\pmb{\pi}^{E}, \pmb{\rho}^{E})$,  the experiment results show that our algorithm can recover $(\pmb{\pi}^{E}, \pmb{\rho}^{E})$. There is no need to let (any) entropy goes to zero as we just recover a given (fixed) MaxEnt-AMFCE. Indeed our algorithm can recover any AMFCE by using demonstrations from this AMFCE, independent of whether it has a maximum entropy or not.
>
>
>
> 2. **The second part**: From Theorem 2, the recovered policy $\pmb{\pi}$ is derived by solving
>
> $$\arg\max_{\pmb{\pi}}\min_{r}L(\pmb{\pi}^{E}, \pmb{\rho}^{E}, \lambda_{\pmb{\pi}}^{*},r)=
> \arg\max_{\pmb{\pi}}\min_{r}\mathbb{E}[\sum_{t=0}^{T}\gamma^{t}r(s_t, a_t, \mu_t)]-J(\pmb{\pi}^{E}, \pmb{\rho}^{E}) - \alpha \sum_{t=0}^{T}\mathbb{E}H(\pi_{t}^{E}|s_t, z_t) \\
> =\arg\max_{\pmb{\pi}}\min_{r}\mathbb{E}[\sum_{t=0}^{T}\gamma^{t}r(s_t, a_t, \mu_t)]-J(\pmb{\pi}^{E}, \pmb{\rho}^{E})\quad (a)$$
> where the first expectation in (a) is taken with respect to $z_t\sim\rho_t^{E}(\cdot)$, $s_t\sim P(\cdot|s_{t-1}, a_{t-1}, \mu_{t-1})$, $a_t\sim\pi_t(\cdot|s_t, z_t)$, $\mu_t=\Phi(\mu_{t-1}, \pi_{t-1}^E, z_{t-1})$ and the second expectation in (a) is taken with respect to $z_t\sim\rho_t^{E}(\cdot)$, $a_t\sim\pi_t^E(\cdot|s_t, z_t)$, $s_t\sim P(\cdot|s_{t-1}, a_{t-1}, \mu_{t-1})$, $\mu_t=\Phi(\mu_{t-1}, \pi_{t-1}^E, z_{t-1})$. The third expectation in (a) is taken with respect to $z_t\sim\rho_t^{E}(\cdot)$, $s_t\sim P(\cdot|s_{t-1}, a_{t-1}, \mu_{t-1})$, $a_t\sim\pi_t(\cdot|s_t, z_t)$, $\mu_t=\Phi(\mu_{t-1}, \pi_{t-1}^E, z_{t-1})$. Therefore, $\alpha \sum_{t=0}^{T}\mathbb{E}H(\pi_{t}^{E}|s_t, z_t)$ does not play a role in the optimization problem over $\pmb{\pi}$. Hence the second equality of (a) follows. This implies that the value of $\alpha$ has no effect on the recovered policy $\pmb{\pi}$. If the entropy regularization value $\alpha$ goes to zero, our algorithm still recovers $\pmb{\pi}^{E}$ from which we collect demonstrations. And the experiment results show that our algorithm can also recover $\pmb{\rho}^{E}$ when $\alpha$ goes to 0.

---

> ### Author Response · Authors · 2022-12-06
> **Dear reviewer a1Rt: we would like to know if you have other questions after our response**
>
> Dear reviewer a1Rt
>
> Thank you very much for your helpful feedback and suggestions, they helped us to improve the paper. We tried to carefully address all of your comments in our response and the updated paper. Please let us know if you have any further questions, and we are very happy to follow up!
>
> Thank you for your time!

---

### Official Review · Reviewer_bLhC · 2022-10-24

**Confidence:** 4
**Correctness:** 2
**Technical Novelty And Significance:** 3
**Empirical Novelty And Significance:** 3
**Recommendation:** 5

**Clarity, Quality, Novelty And Reproducibility:**

Although most of the high-level explanations are quite clear, several important parts of the paper need clarification before I can claim that I am confident the definitions and the results are correct.

**Strength And Weaknesses:**

As mentioned above, the main contribution is the IRL method for correlated equilibria in MFG. There are relatively few papers addressing the question of IRL for MFG so it is interesting to foster our understanding of this problem.

However, I am not sure if the notion of time-dependent correlated equilibrium in MFG is the first one, as claimed by the authors (it seems that some relevant references should be added on this point), and there are several statements and proofs that I could not fully understand.

Some detailed comments:
(1) “this paper is the first focusing on MFCE with the correlation device providing time-dependent recommendations and allowing adaptive belief updates for individual agents.”
Here I would appreciate a more detailed comparison with the following references:
- (Campi & Fischer 2022): their model is also set in finite horizon and it seems to me that their Remark 3.1 explains how to see the recommendation as being time-dependent
- Muller, P., Elie, R., Rowland, M., Lauriere, M., Perolat, J., Perrin, S., Geist, M., Piliouras, G., Pietquin, O. and Tuyls, K., 2022. Learning Correlated Equilibria in Mean-Field Games. arXiv preprint arXiv:2208.10138: their model seems to cover both static and finite horizon problems

(2) Example 1: It seems that the problem is actually static. It would be nice to have a time dependent example, since the authors claim that the time-dependent structure is one of the main features of the proposed model.

(3) Corollary 2: I do not understand the (one-line) proof. It would be useful to spell out the objective and the constraints. My understanding is that there is a constraint on $\mu$ which involves the function $\Phi$ defined in (3). I do not see why this constraint is linear, given that $\Phi$ can depend in a non-linear way on $\mu_t$. Please clarify.

(4) Proposition 2: I am sorry but I do not understand the proof of necessity. Could you please explain where the inequality comes from? First of all, the assumption on $\mathcal{R}$ is a non-strict inequality whereas here there is a strict inequality. Second, the inequality in the assumption is summed over time steps. But here this is time-step per time-step. The sum can be negative without each term being negative. Last, since $a’$ is chosen outside the expectations on $\pi$ and $\rho$, it seems that this reasoning only allows for “global” deviations, i.e., choosing the sequence of actions a priori, before seeing the recommended actions. But in the definition of AMFCE, it seems that we should allow any deviation function $u$, which allows picking the new action after seeing the recommended action. I am not sure how to resolve this and I think that further clarification is required to make sure that the proof is correct.

(5) Equation (8): It is hard to understand the meaning of $L$ because there is $\rho$ in the right-hand side but not in the left-hand side. I am having trouble understanding how $L$ can characterize an AMFCE if it does not depend on $\rho$. I would appreciate more clarifications on this point (see also the next comment).

(6) Proposition 3: From the definition of MFRL, the operator in the left-hand side should return a pair $(\boldsymbol\pi,\boldsymbol\rho)$. But in the right-hand side, the $\argmin$ only returns $\boldsymbol\pi$. More generally it seems that $\boldsymbol\rho$ is frequently omitted although it is a key component of the AMFCE notion, which is quite confusing.

(7) Proposition 3: Definition 5 defines the MaxEnt-AMFCE concept without neural networks. But when it is used in Proposition 3, it involves neural networks. Then it is not clear whether the two notions coincide exactly (I imagine that fixing a neural network architecture implies some changes in the MaxEnt-AMFCE definition).

Typo:
- Between (3) and (4): Should $\pi_i$ be $\pi_t$?
- Page 4: “has an incentive unilaterally deviate” → to
- The notations seem to frequently switch $i$ and $t$ for time, which is a bit confusing. Could you please harmonize the notation?


**Summary Of The Paper:**

This paper studies mean field games (MFG) using a notion of correlated equilibrium and proposes an inverse reinforcement learning (IRL) method. The authors mean field games where a representative agent solves a finite horizon MDP, and globally, we look for a Nash equilibrium. They introduce a notion of correlated equilibrium in this context and prove some properties. Even simple models might have several such solutions. So the authors propose a modification based on a maximum-entropy approach. Then, they introduce an IRL method and implement it on several examples.

**Summary Of The Review:**

Overall, the paper makes an interesting contribution to a relatively little studied question. However, at this stage it seems to me that extra clarifications are needed to assess the novelty and the correctness of the results.

---

> ### Author Response · Authors · 2022-11-18
> **Response: Part 1**
>
> Thank you for your feedback!  We have responded below to each of your comments in detail. All major changes of the submission are marked in red.
> ### Comparisons with (Campi & Fischer 2022) and (Muller et. al. 2022).
>
> First, we want to clarify the main difference between AMFCE and other MFCEs.
> We assume that the recommendation $a_{t}$ at time step $t$ is dependent on $\rho_{t}$, which means that the recommendation $a_t$ cannot be realized until time step $t$.
> - Comparison with (Campi & Fischer 2022):\
>     Campi and Fischer assume that a mediator recommends the same stochastic policy to the entire population. Although the model is simpler, it is less realistic in practice. For many practical applications, the mediator recommends a single action to each agent instead of the entire stochastic policy which may be difficult for the individual to implement. For example, Google Map provides one recommended route to each driver rather a distribution over all possible routes. Please refer to Section 1 for a more detailed explanation.
> - Comparison with (Muller et. al. 2022):\
>     Muller et al. assume that the mediator recommends a deterministic policy sampled from some distribution over the deterministic policy space, and this policy won't change throughout the game. This is equivalent to recommending a sequence of actions sampled from a fixed policy at the beginning of the game. In other words, the recommended actions have been realized at the beginning of the game. Although the finite-horizon case is discussed in a subsection, the theoretical results (such as the existence of the correlated equilibrium) are only established for the infinite horizon and stationary case.
> - In our framework, recommended actions are sampled from Markovian policies that depend on the correlated signal sampled from correlation device $\rho_t$ at every time step. This is a more realistic framework as the correlation device at time t may not be realized/revealed at time 0.
> ### Example 1
> >Example 1: It seems that the problem is actually static. It would be nice to have a time dependent example.
>
> We have updated a new example in Appendix C.1. This example is a finite horizon game and we provide an AMFCE equilibrium (in explicit form). In addition, we hope that the comparisons with (Campi & Fischer 2022) and (Muller et. al. 2022) above could partially answers this question as well.
> ### Corollary 2
> >I do not see why this constraint is linear, given that $\Phi$ can depend in a non-linear way on $\mu_{t}$. Please clarify.
>
> - We apologize for the confusion. The key idea for the uniqueness is not utilizing the linear constraint but rather the convexity of the $\Delta$ function. Namely, the uniqueness holds if $\Delta(\pmb{\pi}, \pmb{\rho})$ is convex w.r.t. $(\pmb{\pi}, \pmb{\rho})$. We have rephrased the intuition before the statement of Corollary 2. The detailed proof is deferred to Appendix D.4.
> - In addition, we want to emphasize that the uniqueness is not necessary for our algorithm design. Any AMFCE could be recovered by using our IL framework.

---

> ### Author Response · Authors · 2022-11-18
> **Response: Part 2**
>
> ### Proposition 2
> > I don't understand the proof of necessity.
>
> Apologies for the confusion. We have polished the proof for better exposition. Below are some detailed explanations.
>
> > The assumption on $\mathcal{R}$ is a non-strict inequality whereas here there is a strict inequality.
>
> In the proof of the necessary condition, we assume that $(\pmb{\pi}, \pmb{\rho})$ is **not** an AMFCE soluton and then  derive a **contradiction** conditon. The single-agent side condition requires that $\Delta_{t}(s, \mu, u; \pmb{\pi}, \pmb{\rho})=\mathbb{E}[Q_{t}^{\pmb{\pi}}(s, u(a), \mu, z)-Q_{t}^{\pmb{\pi}}(s, a, \mu, z)]\le0$. As $(\pmb{\pi}, \pmb{\rho})$ is not an AMFCE, there exists a time step $t\in \mathcal{T}$ such that $\Delta_{t}(s, \mu, u; \pmb{\pi}, \pmb{\rho})=\mathbb{E}[Q_{t}^{\pmb{\pi}}(s, u(a), \mu, z)-Q_{t}^{\pmb{\pi}}(s, a, \mu, z)]>0$ (This inequality is strict).
> > The inequality in the assumption is summed over time steps. But here this is time-step per time-step. The sum can be negative without each term being negative.
>
> In this proof, we want to derive a **contradiction** by showing that the sum $\mathcal{R}(a_{0:T}, \pmb{\pi}, \pmb{\rho})$ is positive if (at least) one term $\Delta_{t}(s, \mu, u; \pmb{\pi}, \pmb{\rho})$ is positive.
>
> Proposition 2 requires that $\mathcal{R}(a_{0:T}, \pmb{\pi}, \pmb{\rho})\leq 0$ holds for any action sequence $a_{0:T}$. If there exists $\Delta_{t}(s, \mu, u; \pmb{\pi}, \pmb{\rho})=\mathbb{E}[Q_{t}^{\pmb{\pi}}(s, u(a), \mu, z)-Q_{t}^{\pmb{\pi}}(s, a, \mu, z)]>0$, the agent can achieve a strictly higher expected return if she chooses action $u(a)$ when she is recommended action $a$ at time step $t$, i.e. $\mathbb{E}\left[\sum_{t=0}^{T} \gamma^t r(s_t, a_t, \mu_t)\right]>J(\pmb{\pi}, \pmb{\rho})$. It implies that there exists an action sequence such that $\mathcal{R}(a_{0:T}, \pmb{\pi}, \pmb{\rho})=\mathbb{E}\left[\sum_{t=0}^{T} \gamma^t r(s_t, a_t, \mu_t)\right]-J(\pmb{\pi}, \pmb{\rho})>0$, which conflicts the assumption.
>
> > In the definition of AMFCE, it seems that we should allow any deviation function $u$, which allows picking the new action after seeing the recommended action.
>
> We apologize for the confusion. We have replaced the 'global deviation' $a'$ with $u(a)$. This doesn't affect the proof. Please also refer to our explanation for the second question about Proposition 2.
> ### Equation (8)
> > There is $\pmb{\rho}$ in the right-hand side but not in the left-hand side.
>
> Thank you for pointing out this typo and we have corrected it.
> ### Proposition 3
> 1. >From the definition of MFRL, the operator in the left-hand side should return a pair $(\pmb{\pi}, \pmb{\rho})$. But in the right-hand side, the $\mathop{\arg\min}$ only returns $\pmb{\pi}$.
>
>     We apologize for the confusion. The MFRL does not return the pair $(\pmb{\pi}, \pmb{\rho})$. It returns $\pmb{\pi}$ given a correlation device $\pmb{\rho}$. We have added a comment to clarify this.
>     Simply applying GAIL to solving AMFCE cannot recover the expert correlation device $\pmb{\rho}^{E}$. This is because MFRL only returns the recovered policy $\pmb{\pi}$. Therefore,  we recover $\pmb{\rho}^{E}$ using the gradient descent method in Proposition 4.
> 2. >Definition 5 defines the MaxEnt-AMFCE concept without neural networks. But when it is used in Proposition 3, it involves neural networks.
>
>     In Proposition 3, we derive the objective in a mini-max form to recover the AMFCE. This inspires us to use GAN to solve the problem in practice. In theory, we assume that neural networks have the capacity to approximate reward. The two notions coincide under this assumption. We leave the complete theoretical development (on what neural network architecture has the desired approximation power) as a future direction. We have added a comment in the submission to clarify this.

---

> ### Author Response · Authors · 2022-12-06
> **Dear reviewer bLhC: we would like to know if you have other questions after our response**
>
> Dear reviewer bLhC
>
> Thank you very much for your helpful feedback and suggestions, they helped us to improve the paper. We tried to carefully address all of your comments in our response and the updated paper. Please let us know if you have any further questions, and we are very happy to follow up!
>
> Thank you for your time!

---

### Official Review · Reviewer_V9BL · 2022-10-25

**Confidence:** 2
**Clarity, Quality, Novelty And Reproducibility:** See above.
**Correctness:** 3
**Technical Novelty And Significance:** 3
**Empirical Novelty And Significance:** 2
**Recommendation:** 6

**Strength And Weaknesses:**

**Strengths:**


Mean field games, correlated eq., and generatie adversarial imitation learning are all relevant and interesting areas, and the authors combine them.
The authors propose quite a collection of non-trivial mathematical results about this combination.
Frankly I did not have time to check the details of the proofs, but the authors kind of seem to know what they are doing, and the high level structure of the proof steps for the existence results (Thm 1) using Kakutani make sense.


**Weaknesses and improvement points:**


There are quite a lot of technical elements combined in this work,
in particular about "foward" MF game theory; and then comes GAIL which is already in itself a non-trivial topic.
This is no principle problem, but makes  it a bit overwhelming to read and to understand details.

One thing I found a bit confusing: are the IL aspects also an original contribution or not? Because they are not listed under the bullet points in Sec. 1. I guess it's original contribution in the sense of the first formulation of the GAIL principle for this mean field game setting?

The writing overall is OKish, but clarity of details needs to be improved, in particular the introductions of the basic concepts and definitions:
* Where is state defined? Is the state space finite?
* I appreciate that the reader is given intuitions about the concepts (e.g., below Def 1). However, I think a bit more mathematical precision would be good here. E.g., how exactly is the Law(s) defined (mathematical expression). I guess there may even be some conditions such that there is such a distro over states from a set/distro of agents.
* A bit confusing that in  Sec. 3, pi does not seem to be the agents policy anymore, but instead part of the recommendation mechanism. (Instead the swap function is sort of the agents's policy now.)

In terms of motivation, I allocate this paper more on the abstract mathematical side than the real-application-motivated side. This is OK - otfen done - but nonetheless there are some motivational weaknesses from my point of view: E.g., the maximum entropy regularier is often used, but frankly I think there is only a vague principled motivation, while the main motivation is usually that it can be efficiently calculated or such things. I'm completely missing the motivation why rational or somewhat rational agents (which is the underlying assumption with game theory) should perform entropy-based equilibrium selection.

Overall, I'm not an expert on mean field games; from a high level the work seems to make sense, though I cannot fully verify the originality of the contributions there; and also not the full technical soundness.

**Summary Of The Paper:**

This work proposes a variant of mean field games, where there is a correlation device, and subsequently introduce the corresponding equilibrium concept.

It analyzes this type of game/equilibrium concept, in particular an existence result (Thm 1).

Additionally, the imitation learning objective for learning the policies (rewards) is proposed, based on translating GAIL to this setting.

A limited number of experiments are conducted.

**Summary Of The Review:**

Technical paper combining interesting areas, with several mathematical contributions; in terms of correctness of proofs, the authors seem to know what they are doing, but I did not check details of the proofs.

Limits and weaknesses are partially in terms of writing and complexity of the overall paper, as well as motivation of some concepts/assumptions.

---

> ### Author Response · Authors · 2022-11-18
> **Response**
>
> Thanks for your positive feedback! We have responded below to each of your comments in detail. All major changes of the submission are marked in red.
>
> >Are the IL aspects also an original contribution or not?
>
> There are existing works on inverse reinforcement learning (IRL) for Mean-field Games (MFG) (https://arxiv.org/abs/2202.06401, https://openreview.net/forum?id=HktK4BeCZ)
>
> However, our paper propose the first imitation learning (IL) framework for Correlated MFG. As explained in the motivating applications (i.e., the traffic routing example and e-commerce market example), many real-world stochastic systems (with interventions from the platform manager) fit into the definition of correlated MFG but not the classic MFG. Therefore, we believe the IRL framwork for Correlated MFG will have broader impacts in real-world applications.
>
> Indeed IRL is a subclass of IL. IRL and IL are different in the sense that whether or not the algorithm recovers the reward function. IRL recovers the reward function and then extracts the policy, while IL extracts the policy directly. In this sense, the IL aspect is also an original contribution.
>
> >Where is state defined? Is the state space finite?
>
> The transition kernel for the state dynamics is defined at the beginning of Section 2. The state space is finite and we have emphasized this at the beginning of Section 2.
>
> >How exactly is the Law(s) defined (mathematical expression).
>
> ${\rm Law}(s_t)$ is the (unconditional) distribution of $s_t$, namely the state distribution of the representative agent. It can be derived by using Kolmogorov equation. Following your comment, we have remoted this notation and added the mathematical expressions in both Definition 1 and Definiton 4. In particular, Definition 1 has been rephrased as the following:
>
> [Definition 1] In (LMFG), a player-population profile ($\pmb{\pi}^\star$, $\pmb{\mu}^\star$)
>   is called a MFNE (under initial state $\mu_0$) if
> - (Single player side) For any policy $\pmb{\pi}$, any time index $t\in \mathcal{T}$, and any initial state $s\in \mathcal{S}$,  $V_t\left(s,\pmb{\pi}^\star,{\pmb{\mu}^\star}\right)\geq V_t\left(s,\pmb{\pi},\pmb{\mu}^\star\right).$
> -  (Population side) $\mu_t^*$ satisfies $\mu^*_{t}(\cdot) = \sum_{s\in\mathcal{S},a\in \mathcal{A}} P(\cdot|s,a,\mu_{t-1}^*)\pi^*_{t-1}(a|s)\mu^*_{t-1}(s)$ with initial condition $\mu^*_0=\mu_0$, for all $t=0, 1, \cdots T$.
>
> The single player side condition captures the optimality of $\pmb{\pi}^\star$, when the population side is fixed. The population side condition ensures the ''consistency'' of the solution: it guarantees that the state  distribution flow of the single player  matches the population state  sequence $\pmb{\mu}^{\star}:=${$\mu_t^\star$}$_{t=0}^{T}$.
>
> The definition of Law($s_t$) in Definition 4 is modified accordingly.
> >A bit confusing that in Sec. 3, pi does not seem to be the agents policy anymore, but instead part of the recommendation mechanism. (Instead the swap function is sort of the agents's policy now.)
>
> - $\pi$ is the distribution from which private recommendations are sampled. Agents can deviate from the recommendations and choose alternative actions (that would lead to the best value function under their current belief). This is why the swap function is defined. When correlated equilibrium is reached, $\pi$ becomes individual agent's policy as no one has the incentive to deviate from this policy $\pi$.
> - By defining the swap function, we established a more general framework with correlated equilibrium, which includes the classical Nash equilibrium as a special case.
> >The motivation why rational or somewhat rational agents (which is the underlying assumption with game theory) should perform entropy-based equilibrium selection?
>
> - The rationality is implicitly assumed in the single-agent side condition in Definition 4. It requires that the agent chooses the policy to maximize the expected return.
> - There might be many adaptive mean-field correlated equilibria (AMFCE) under the proposed model. Each AMFCE solution satisfies both conditions in Definition 4 (hence the single agent side condition). Therefore, agents are rational in all AMFCE solutions. In particular, the MaxEnt-AMFCE is a subclass of AMFCE.
> - Indeed, the equilibrium selection principle based on maximum entropy is widely employed in the field of game theory (please refer to http://arxiv.org/abs/1308.3506, https://www.cs.cmu.edu/~bziebart/publications/maxcausalent-correlated.pdf). This is because equilibrium with bigger entropy is expected to be more stable in complex stochastic systems and have better exploration structure.
> - We assume that the equilibrium is MaxEnt-AMFCE due to the above mentioned properties of the maximum entropy. We want to clarify that our algorithm can recover any AMFCE (which may not have the maximum entropy). Apologies for the confusion and we have added a comment to clarify this in Section 4.1.

---

> ### Author Response · Authors · 2022-12-06
> **Dear reviewer V9BL: we would like to know if you have other questions after our response**
>
> Dear reviewer V9BL
>
>    Thank you very much for your helpful feedback and suggestions, they helped us to improve the paper. We tried to carefully address all of your comments in our response and the updated paper. Please let us know if you have any further questions, and we are very happy to follow up!
>
> Thank you for your time!

---

### Decision · Program_Chairs · 2023-01-20

**Decision:**

Reject

**Justification For Why Not Higher Score:**

NA

**Justification For Why Not Lower Score:**

NA

**Metareview: Summary, Strengths And Weaknesses:**

I skimmed through the paper myself. This paper proposes to study imitation learning in mean-field games but with a generalized version of game equilibrium that allows for correlated equilibria. While the mathematical concept is clearly defined and sufficiently discussed in the paper, I'm having trouble appreciating the motivation to study such a notion in practice. In multi-agent systems, it seems very unlikely that existing agents are acting according to an equilibrium policy. In fact, they are often far from optimal. For instance, in both online pricing and traffic navigation examples, I don't think the behavior of human users is anywhere near the equilibrium policy. This is not surprising because, in general, solving the equilibrium of Markov games is computationally intractable. Given that such a scenario almost never appears in practice, the motivation for studying such a setting is questionable.

For the above reasons, I vote for the rejection of this paper.

**Summary Of Ac-Reviewer Meeting:**

Unfortunately, I'm not able to reach the reviewers.